# Nitrapyrin Mitigates Nitrous Oxide Emissions, and Improves Maize Yield and Nitrogen Efficiency under Waterlogged Field

**DOI:** 10.3390/plants11151983

**Published:** 2022-07-30

**Authors:** Baizhao Ren, Zhentao Ma, Bin Zhao, Peng Liu, Jiwang Zhang

**Affiliations:** State Key Laboratory of Crop Biology, College of Agronomy, Shandong Agricultural University, Taian 271018, China; renbaizhao@sina.com (B.R.); sdmazht@163.com (Z.M.); zhaobin1981824@163.com (B.Z.); liup@sdau.edu.cn (P.L.)

**Keywords:** summer maize, waterlogging, nitrous oxide, nitrapyrin, nitrogen use efficiency

## Abstract

In order to explore the effects of nitrapyrin (N-Serve) application on greenhouse gas emission and nitrogen (N) leaching of a waterlogged maize (*Zea mays* L.) field, we investigated the effects of applying nitrapyrin on soil ammonium (NH_4_^+^-N) and nitrate nitrogen (NO_3_^−^-N) content, nitrous oxide (N_2_O) fluxes, and the warming potential (GWP_N_2_O_) in a waterlogged maize field. The design included three treatments: waterlogging treatment with only urea application (V-3WL), waterlogging treatment with urea and nitrapyrin application (V-3WL+N), and no waterlogging treatment applying only urea (CK). Our results revealed that waterlogging led to the increase of nitrate concentrations across the soil profile, thus potentially increasing N leaching and decreasing N use efficiency. The accumulated N_2_O emissions increased significantly in waterlogged plots compared to control plots, and maximum N_2_O emission fluxes occurred during the process of soil drying after waterlogging; this resulted in an increase in GWP_N_2_O_ and N_2_O greenhouse gas intensity (GHGI_N_2_O_) by 299% and 504%, respectively, compared to those of CK. However, nitrapyrin application was able to reduce N_2_O emissions. Nitrapyrin application was also good for decreasing GWP_N_2_O_ and GHGI_N_2_O_ by 34% and 50%, respectively, compared to V-3WL. In addition, nitrapyrin application was conducive to reduce N leaching and improve N use efficiency, resulting in a yield increase by 34%, compared to that of V-3WL. The application of nitrapyrin helped to mitigate agriculture-source greenhouse effects and N leaching induced by waterlogging, and was a high N-efficient fertilizer method for a waterlogged field.

## 1. Introduction

Agriculture has become a major source of global greenhouse gas emissions (GHGs), which must be substantially reduced to minimize the impacts of climate change [1]. Agriculture accounts for 10%–20% of global GHGs produced by human activities, with nitrous oxide (N_2_O) accounting for 60% of total agricultural emissions [2,3]. In China, the GHGs produced by the agriculture sector are primarily N_2_O, comprising 15% of the total GHGs of the country [2]. Nitrous oxide, a long-lived greenhouse gas, contributes to global warming and also serves as an atmospheric tracer of anthropogenic changes to the global N cycle, with a global warming potential 273 times that of CO_2_ [4,5,6].

In agricultural fields, the N_2_O emissions are mainly produced by the chemical and organic N inputs [7]. Because of respiration by soil microbes, soil animals, and plant roots after nitrogen (N) fertilization, soil becomes a significant source of N_2_O emissions [8,9]. The biomass, physiology, and biochemistry of soil microbes are affected by temperature, water content, organic content, pH, redox potential, and the texture of the soil, among other factors; in turn, this affects the rate of soil GHGs [10,11]. Global climate change is predicted to increase the frequency and intensity of extreme precipitation events [12,13,14,15], which could dramatically alter soil GHGs. For example, intensified precipitation regimes would lead to higher incidences of soil waterlogging or flooding and the changes of soil hydrological cycles [16]. N_2_O emission rates depend on the interaction among soil types, climate, and farm management, which influence soil microbial processes and the diffusion of gaseous N_2_O to the atmosphere [17]. Humid tropical soils are generally associated with the production of large amounts of gaseous N oxides, including N_2_O [18].

Excessive rainfall or irrigation can also lead to high rates of nitrate (NO_3_^−^) leaching from soils, resulting in the losses of reactive N from maize fields. Waterlogging also restricts nutrient absorption and use by maize roots, leading to a substantial reduction of N efficiency [19]. Nitrification inhibitors can effectively suppress the oxidation of ammonium (NH_4_^+^) to NO_3_^−^, thus reducing N loss from soils and improving N uptake by crops [20]. The companion of nitrification inhibitors with N fertilizer has been applied as an agronomic practice to reduce N leaching and lessen N_2_O and NO emissions [20,21]. Nitrapyrin (2-chloro-6-(trichloromethyl) pyridine) is similar to other customarily used nitrification inhibitors, such as dicyandiamide (DCD) and 3,4-dimethylpyrazole phosphate (DMPP), and has been customarily applied to crop fields to inhibit NO_3_^−^ leaching with great success [22,23,24,25]. The application of nitrapyrin can reduce N leaching by inhibiting the release of soil N_2_O by 49% and 24%, respectively, compared with urea and urea-ammonium nitrate application during the maize growth period [26,27]. This suppresses the oxidation of NH_4_^+^ to NO_3_^−^ and maintains N in the form of NH_4_^+^ in soil for a longer period [28]. The effectiveness of nitrification inhibitors on N_2_O emissions depends on environmental parameters, such as waterlogging, heat, cold, and so on [21,29,30,31]. Nitrification inhibitors affect N_2_O emissions more effectively under higher soil moisture levels by regulating the abundance of denitrifying genes (narG, nirK, and nosZ) [32,33,34]. Our previous studies revealed that urea combined with nitrapyrin increased the NH_4_^+^ content in the soil, prolonged the fertilizer efficiency, and promoted the absorption and metabolism of N in maize, thereby promoting the recovery of maize and alleviating the reduction of plant dry matter accumulation caused by waterlogging [35]. Moreover, the application of nitrapyrin can lead to higher yields in waterlogged summer maize by optimizing the absorption and relocation of N, which effectively improves N use efficiency (NUE) and the N harvest index [35,36].

However, little research has been conducted on the role of nitrapyrin in mitigating the effects of waterlogging on GHGs and N leaching. In this study, we conducted a field experiment to measure the effects of nitrapyrin application and waterlogging on N_2_O emissions, and on the content of soil NO_3_^−^-N and NH_4_^+^-N in a maize field. The results of this study will help in developing a strategy to reduce GHGs and N leaching in waterlogged maize.

## 2. Materials and Methods

### 2.1. Plant Materials and Experimental Location

A field experiment was conducted at the experimental farm (36°10′ N, 117°04′ E, 151 m a.s.l.) maintained by the State Key Laboratory of Crop Biology of Shandong Agricultural University, Taian, China in 2016 and 2017. This region was characterized by a temperate continental monsoonal climate with a mean annual temperature of approximately 13 °C, a frost-free period of 195 days, and annual precipitation of 697 mm. The rainfall in two maize growing seasons were mainly concentrated in July, accounting for 46.0% and 54.8%, respectively. The 0–20 cm top-soil of the experimental field consisted of brown loam, which contained 10.7 g kg^−1^ organic matter, 0.9 g kg^−1^ total N, 50.7 mg kg^−1^ available phosphorus (molybdenum-antimony [Mo-Sb] colorimetry), and 86.2 mg kg^−1^ available potassium (Flame photometry). Denghai605 (DH605), a commonly grown maize (*Zea mays* L.) hybrid, was used for this experiment. Maize seeds were sown on June 16 at a density of 67,500 plants ha^−1^.

### 2.2. Experimental Design

Each plot measured 4 × 4 m^2^ and was surrounded by four 4 × 2.3 m^2^ polyvinyl chloride (PVC) boards, which acted as water barriers. Each PVC board was buried 2.0 m below the soil surface, with the remaining 0.3 m above ground. In the waterlogged plots, the water level was maintained at 2~3 cm above the soil surface for 6 days, starting when maize plants were at the third leaf stage (V3). After 6 days, all the water was drained from the soil surface. Two treatments were tested in this experiment: a waterlogging treatment with urea application only (V-3WL); and a waterlogging treatment with urea and nitrapyrin application (V-3WL+N). Control plots (CK) were not waterlogged, but applied only urea. Each treatment had three replicates, and treatments were randomly applied to plots in the field. Fertilizer was applied: 210 kg ha^−1^ N (urea with 46% N); 84 kg ha^−1^ phosphorus pentoxide (P_2_O_5_; calcium superphosphate with 17% P_2_O_5_); and 168 kg ha^−1^ potassium oxide (K_2_O; muriate of potash with 60% K_2_O) at the beginning of the experiment. For nitrapyrin treatment, 2550 mL ha^−1^ nitrapyrin was mixed uniformly with urea and incorporated into the soil via ploughing. The rate of nitrapyrin was 0.24% of the rate of urea-N application.

### 2.3. Soil N_2_O Fluxes Measurements

Soil N_2_O fluxes were estimated using a static-chamber method [37]. These gas fluxes were measured between 8:00 am and 11:00 am daily from the first day of waterlogging to the last day of soil drying using closed-chamber every other day. The closed chamber (length 0.35 m × width 0.35 m × height 0.2 m) was enclosed by plastic sheets. The exterior of the chamber was insulated with heat-insulating cystosepiment (0.5 cm thick foam board) to prevent temperature changes, and an air vent was installed in the middle of the chamber. A pedestal was placed under the chamber, and the base was sealed using water to ensure that the external environment did not affect the interior of chamber when gas samples were collected. Gas samples (50 mL) were collected using glass syringes from the chamber headspace at 0, 10, 20, and 30 min after placing the chamber on the soil. Concentrations of N_2_O in the gas samples were detected using an Agilent GC7890 gas chromatograph (Agilent, Santa Clara, CA, USA) equipped with an electron capture detector (ECD). N_2_O flux was calculated as:(1)$$J=\frac{\mathrm{dc}}{\mathrm{dt}}\times \frac{\mathrm{MP}}{V0}\times \frac{T0H}{P0T}$$
where J is flux (mg m^−^^2^ h^−1^), and dc/dt is the change in gas concentration (c, mg m^−3^) against time (t, hour). M is the molar mass (mg mol^−1^) of each gas, P is atmospheric pressure (KPa), T is the absolute temperature (K) during sampling, H is the height (m) of headspace in chamber, and V0, T0, and P0 are the gas molar volume (m^3^ mol^−1^), absolute air temperature (K), and atmospheric pressure (KPa), respectively, under standard conditions.

N_2_O warming potential (GWP_N_2_O_, kg CO_2_-eq m^−2^) was calculated by multiplying the N_2_O emission fluxes by radiative forcing potentials. The equation is as follows [4]:(2)$${\mathrm{GWP}}_{N_{2}O}=f_{N_{2}O}\times 273$$
where f_N_2_O_ is N_2_O emission flux.

N_2_O greenhouse gas intensity (GHGI_N_2_O_, kg kg^−1^) represented the comprehensive greenhouse effect of each treatment and was calculated as follows [38,39]:(3)$${\mathrm{GHGI}}_{N_{2}O}=\frac{{\mathrm{GWP}}_{N_{2}O}}{Y}$$
where Y (kg ha^−1^) is the grain yield of summer maize for each treatment.

### 2.4. Soil NH_4_^+^-N and NO_3_-N Content

The soil samples were divided into three layers from 0 to 90 cm, each one with a height of 30 cm. The soil sample of each layer was placed by an earth drill into a Ziploc bag at the sixth leaf stage (V6), tasseling stage (VT), and physiological maturity stage (R6) [37]. Soil NH_4_^+^-N and NO_3_^−^-N were extracted with 1 *M* KCl, and filtered through a 0.45-μm membrane filter to remove insoluble particulates. The content of the soil NH_4_^+^-N and NO_3_^−^-N were measured by the AA3 Continuous Flow Analytical System [40]. Three replicate soil samples were collected in each treatment.

### 2.5. Nitrogen Efficiency and N Budget

Five representative plant samples were obtained from each plot at the physiological maturity stage (R6). The samples were dried at 80 °C in a force-draft oven (DHG-9420A, Bilon Instruments Co. Ltd., Shanghai, China) to a constant weight and weighed separately. The total N was measured using the Kjedahl method. Nitrogen use efficiency (NUE, kg kg^−1^), N partial factor productivity (NPFP, kg kg^−1^), N harvest index (NHI, %), and the apparent N budget (kg ha^−1^) were calculated to investigate the performance of agricultural management practices, using the following equations:(4)$$\mathrm{NPFP}=\frac{Y}{\mathrm{NA}}$$
(5)$$\mathrm{NUE}=\frac{Y}{\mathrm{TN}}$$
(6)$$\mathrm{NHI}=\frac{\mathrm{GN}}{\mathrm{TN}}$$
(7)Apparent N budget=ΔN+NA−TN
where NA (kg N ha^−1^) is N applied, TN (kg ha^−1^) is the total N uptake by plant, GN (kg ha^−1^) is the grain N amount, and ∆N (kg ha^−1^) is the change in the soil inorganic N (including NH_4_^+^-N and NO_3_^−^-N) before and after maize planting.

### 2.6. Crop Yield

To determine the maize yield and ear traits, 30 ears were harvested at the physiological maturity stage (R6) from three rows at the center of each plot. All the kernels were air-dried, and the grain yield was measured at 14% moisture, the standard moisture content of maize in storage or for sale in China (GB/T 29890-2013).

### 2.7. Data Analysis

Analysis of variance (ANOVA) was performed according to the general linear model procedure of SPSS (Ver. 17.0, SPSS, Chicago, IL, USA). The least significant difference (LSD) between the means was estimated at the 95% confidence level. Unless otherwise indicated, significant differences are at *p* ≤ 0.05. LSD; this was used to compare the adjacent means arranged in order of magnitude.

## 3. Results

### 3.1. Grain Yield

Waterlogging significantly decreased grain yield. The grain yield of V-3WL was 31% lower than that of CK across years. However, the application of nitrapyrin was beneficial to increase yields in waterlogged plots, with the grain yield being 34% higher in the V-3WL+N treatment than that in the V-3WL treatment. In addition, the kernel number and 1000-grain weight were significantly increased by 19 and 9% for V-3WL+N, respectively, compared to those of V-3WL across years (Table 1).

### 3.2. N_2_O Emissions

The three treatments during the summer maize season showed significant temporal variation in N_2_O emissions. N_2_O emission increased significantly after waterlogging, and the maximum N_2_O was recorded during the process of soil drying. The N_2_O emission fluxes in CK, V-3WL+N, and V-3WL ranged from 6.9 to 221.1, 15.8 to 281.0, and 18.6 to 370.0 μg m^−2^ h^−1^, respectively. The variation trends of N_2_O fluxes in the two waterlogging treatments were similar, while the treatment with nitrapyrin decreased significantly. After waterlogging, the cumulative emission flux of N_2_O increased significantly, showing a trend of V-3WL>V-3WL+N>CK (Figure 1).

Likewise, waterlogging significantly increased the GWP_N_2_O_ and GHGI_N_2_O_. However, after the addition of nitrapyrin, the GWP_N_2_O_ and GHGI_N_2_O_ were significantly decreased, and V-3WL+N decreased by 34% and 50%, respectively, compared with the V-3WL treatment (Figure 2).

### 3.3. Soil NO_3_^−^-N and NH_4_^+^-N Concentrations

The soil N was rapidly leached in the waterlogged soil due to high levels of soil moisture. The two-year average results showed that the NO_3_^−^-N concentration in the top (0–30 cm) soil layer in the V-3WL treatment was 40% lower than that in the CK soil at the V6 stage; however, the NO_3_^−^-N concentrations in the mid (30–60 cm) and deep (60–90 cm) soil layers were 22% and 15% higher in the V-3WL treatment, respectively, compared to that in CK. (There was no significant difference between the V-3WL and CK treatments in the 60–90 cm soil layer in 2017.) When nitrapyrin was applied, the transformation of NH_4_^+^-N to NO_3_^−^-N was inhibited. The NO_3_^−^-N concentrations in V-3WL+N treatment increased by 14% in the topsoil layer, and decreased by 12% and 31% in the mid and deep soil layers, respectively, compared to V-3WL. At VT, the NO_3_^−^-N concentration of the waterlogging treatment in the 0–30 cm and 30–60 cm soil layers was lower than CK. In 2017, the NO_3_^−^-N concentration of the V-3WL treatment in each soil layer at the R6 stage was significantly higher than that of V-3WL+N, while there was no significant difference between the two waterlogging treatments in 2016 (Figure 3).

At the V6 stage, the NH_4_^+^-N concentration in the 0–30 cm, 30–60 cm, and 60–90 cm soil layers of the V-3WL+N was significantly increased, which was 77%, 28%, and 54% higher than the V-3WL treatment, respectively. However, at VT, the NH_4_^+^-N concentration in the deep soil layer of the V-3WL treatment was significantly higher than that of the other treatments, which increased by 26% and 27% compared with CK and V-3WL+N, respectively. Obviously, the NH_4_^+^-N concentration of each soil layer at R6 showed a similar trend of V-3WL>V-3WL+N>CK (Figure 4).

V-3WL, waterlogging treatment with urea application only; V-3WL+N, waterlogging treatment with urea and nitrapyrin application; CK, not waterlogged, but applied only urea. The error bars represent 95% confidence intervals. 

### 3.4. Nitrogen Efficiency and N Budget

Nitrogen accumulation was significantly reduced after waterlogging. The total N accumulation was 24% lower than that of CK across years. Moreover, the nitrogen partial factor productivity (NPFP), nitrogen use efficiency (NUE), and nitrogen harvest index (NHI) were 31, 5, and 17% lower than those of CK, respectively. Nitrapyrin application effectively alleviated the reduction of N accumulation and N efficiency induced by waterlogging. The total N accumulation, NPFP, NUE, and NHI of V-3WL+N was increased by 14, 34, 10, and 12% across years, respectively. All the treatments had a surplus of soil N, and the apparent surplus of soil N increased significantly after waterlogging. However, nitrapyrin application reduced the apparent soil N surplus in 2016 (no significant difference in 2017) (Table 2).

## 4. Discussion

The soil N_2_O was largely produced from microbial nitrification and denitrification, which were affected by environmental conditions such as soil temperature, moisture content, organic matter content, and pH [17,41]. Of these factors, soil moisture content had the greatest effect on N_2_O emission [42]. An increase in soil moisture, such as from natural (e.g., rainfall) and artificial (e.g., irrigation) processes, resulted in short-term increases in N_2_O emissions and, thus, lower N acquisition and NUE by crops [35]. Maximum N_2_O flux was reached at 84–86% water-filled pore space (WFPS), which represented the percentage of soil saturation by water. At less than 70% WFPS, soil N_2_O flux, mostly produced by nitrification, increased with increasing soil moisture content and the application of N fertilizer [24]. Conversely, when WFPS was more than 70%, N_2_O was mostly produced by denitrification [43,44,45]. In addition, sufficient mineral N content would promote the release of N_2_O emissions by changing the nitrification and denitrification rate of microorganisms, and even form “hot spots” of N_2_O emissions [4].

Our previous results indicated that waterlogging limited plant growth and lowered grain yield by decreasing both NUE and N fertilizer recovery efficiency in summer maize [35,36]. In this present study, we found that soil N_2_O fluxes were significantly increased after waterlogging (Figure 1). The maximum N_2_O flux was recorded during the process of soil drying, similar to observations by Jie et al. (1997) [46] for paddy fields. This was probably because the water layer covering the soil surface reduced soil permeability and promoted denitrification. When the soil was waterlogged, gaps among soil particles were completely filled with water. Under this condition, N_2_O accumulated in the soil and did not revert back to gaseous N (N_2_). However, N_2_O was released from the soil as the soil gradually dried. Thus, waterlogging led to increased N_2_O emissions from our experimental plots and decreased the NUE of the maize crop. The interaction of local temperature, soil, and other environmental factors affect N_2_O emissions fluxes [47]. Compared with 2016, the peak N_2_O emission of the waterlogging treatment decreased; however, the duration was prolonged, and the cumulative N_2_O emission flux increased in 2017. Multi-day rainfall from June to July in 2017 may have contributed to differences in N_2_O emission fluxes between the two maize seasons. On a broader spatial scale, waterlogging could increase the rate of N_2_O emissions. This would lead to an increased contribution to the greenhouse effect from agricultural activities with a significant increase of GWP_N_2_O_ and GHGI_N_2_O_ by 299% and 504%, respectively, compared to those of CK; this confirms that waterlogging contributed significantly to the greenhouse effect. However, the application of nitrapyrin was shown to improve the absorption and use of N fertilizer and N distribution in grains, which increased the NUE of summer maize grown under waterlogged conditions [35], and enhanced total N accumulation and N fertilizer recovery efficiency (Table 1 and Table 2). Nitrapyrin could reduce N losses through leaching and gas diffusion by inhibiting soil nitrification [32,33]. In this study, the application of nitrapyrin resulted in lowered N_2_O emissions, decreasing GWP_N_2_O_ and GHGI_N_2_O_ by 34% and 50%, respectively. Visibly, nitrapyrin could reduce the effect of waterlogging on soil GHG emissions and help to lower the contribution from agricultural activities to the greenhouse effect.

The leaching of the soil N, which was exacerbated by waterlogging, resulted not only in fertilizer losses, but also in serious environmental problems [48,49]. Soil particles have a negative surface charge (cation exchange capacity); the soil is able to bind positively charged ions (such as NH_4_^+^) and prevent such ions from leaching. In contrast, NO_3_^−^ is a negatively charged ion that does not bind to soil and is easily leached in soil solutions [50]. Our previous study showed that waterlogging limited root growth and development, lowering the ability of the plant to absorb N, and decreased the plant NUE by increasing soil N leaching in the form of NO_3_^−^ [19]. At the early waterlogging stage, the soil moisture content was very high, and the soil N would be leached at high rates. At the V6 stage, the NH_4_^+^-N and NO_3_^−^-N concentrations were lower in the topsoil layers, and higher in the mid and deep soil layers in waterlogged soil compared to non-waterlogged soil (Figure 3). By contrast, the NH_4_^+^-N concentrations in waterlogged soil with nitrapyrin (V-3WL+N treatment) were higher in all the soil layers compared to the concentrations in waterlogged soil without nitrapyrin (V-3WL treatment). However, the NO_3_^−^-N concentrations in the mid and deep soil layers were lower in the V-3WL+N treatment compared to those in the V-3WL treatment (Figure 4). This result might be due to the presence of nitrification inhibitors, which prevented the transformation of NH_4_^+^-N into NO_3_^−^-N. As the soil dried, the soil moisture content returned to baseline levels, and the process of soil N leaching slowed. However, the disorder of root growth and development caused by waterlogging led to a reduced ability to absorb N and the accumulation of N around the root rhizosphere [19]. Therefore, NH_4_^+^-N concentrations in the topsoil layers of waterlogged plots were higher than those of the control plots when the plants were at the VT and R6 stages. These results showed that the N uptake and NUE of summer maize were reduced in waterlogged soils, resulting in an increase in the apparent budget of soil N.

The interannual N_2_O emission fluxes differences did not have a significant impact on the apparent budget of soil N (Table 2). The interannual differences appeared to be offset by soil inorganic N levels before N application in the summer maize season. In addition, when calculating the apparent budget of the soil N in this study, the mineralized and fixed amount of soil N were not included. The results showed that the N fertilization treatments had a surplus of N in summer maize season, and the waterlogging treatments had a greater risk of N loss. Nitrification inhibitors could reduce N leaching rates, and improve N absorption and use efficiency in plants; thus, this mitigates waterlogging damage on N leaching and N use efficiency. Overall, the application of nitrapyrin to waterlogged fields can help to reduce N loss and GHG_N_2_O_ flux, and increase grain yield (Figure 5). Nitrapyrin, an eco-friendly and N-efficient fertilizer companion, was useful for waterlogged soil conditions by helping to decrease N leaching and reducing the agricultural contribution to the greenhouse effect.

## 5. Conclusions

In this study, waterlogging significantly decreased crop yield and increased the cumulative emission flux of N_2_O, warming potential and greenhouse gas intensity. Nitrapyrin application not only helped to increase grain yield and N efficiency of maize grown in waterlogged soil, but also reduced the GHGI_N_2_O_ and GWP_N_2_O_ of waterlogged soil as well as the contribution to the greenhouse effect from agricultural sources.

## Figures and Tables

**Figure 1 plants-11-01983-f001:**
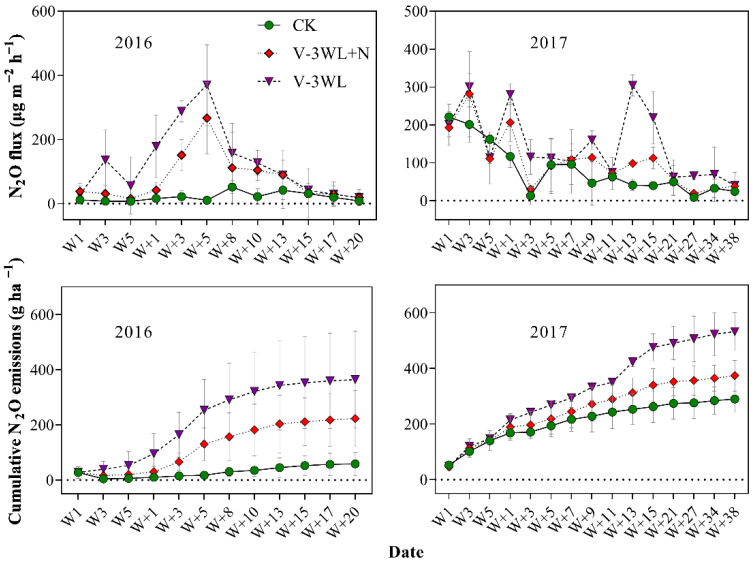
Effects of applying nitrapyrin on the soil N_2_O emission flux under a waterlogged field. V-3WL, waterlogging treatment with urea application only; V-3WL+N, waterlogging treatment with urea and nitrapyrin application; CK, not waterlogged, but applied only urea; Wn, waterlogging duration; W+n, the day after waterlogging. The error bars represent 95% confidence intervals.

**Figure 2 plants-11-01983-f002:**
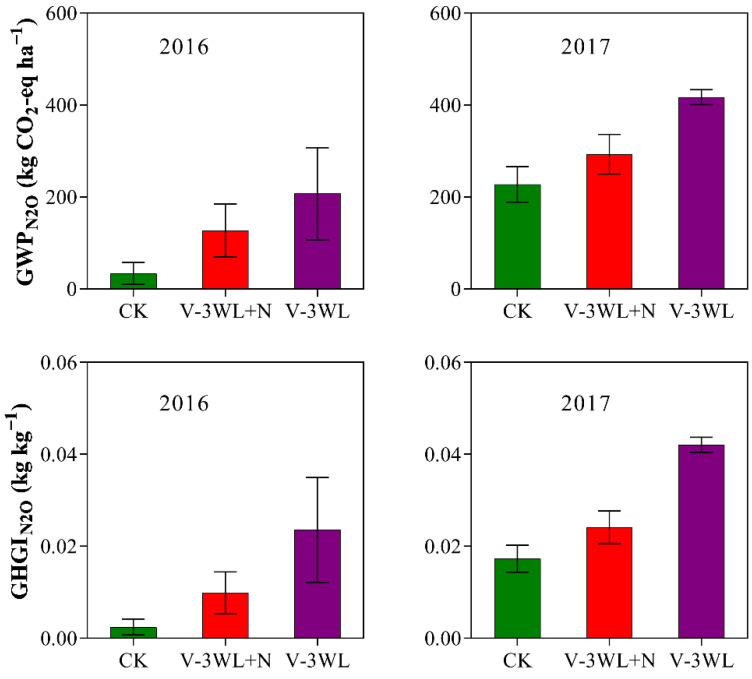
Effects of applying nitrapyrin on the warming potential under a waterlogged field. V-3WL, waterlogging treatment with urea application only; V-3WL+N, waterlogging treatment with urea and nitrapyrin application; CK, not waterlogged, but applied only urea. The error bars represent 95% confidence intervals.

**Figure 3 plants-11-01983-f003:**
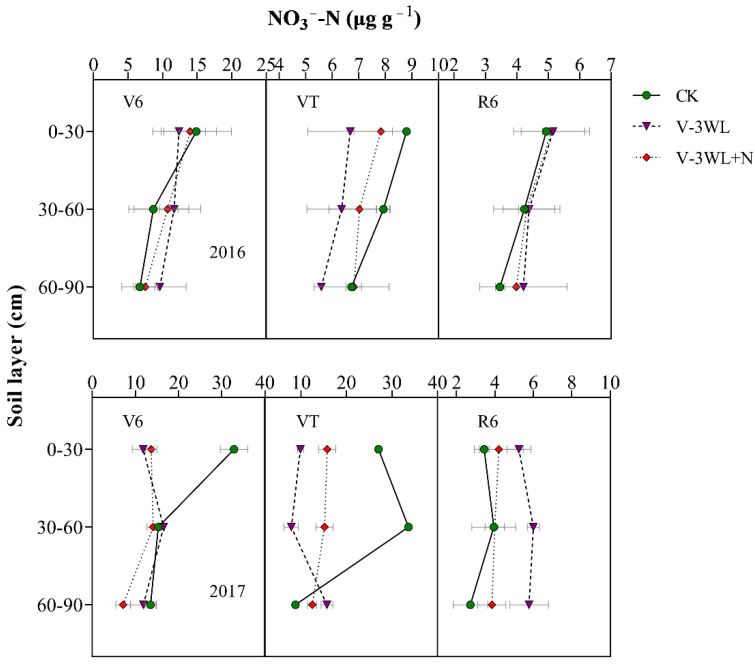
Effects of applying nitrapyrin on the soil NO_3_^−^-N content in a waterlogged maize field. V-3WL, waterlogging treatment with urea application only; V-3WL+N, waterlogging treatment with urea and nitrapyrin application; CK, not waterlogged, but applied only urea. The error bars represent 95% confidence intervals.

**Figure 4 plants-11-01983-f004:**
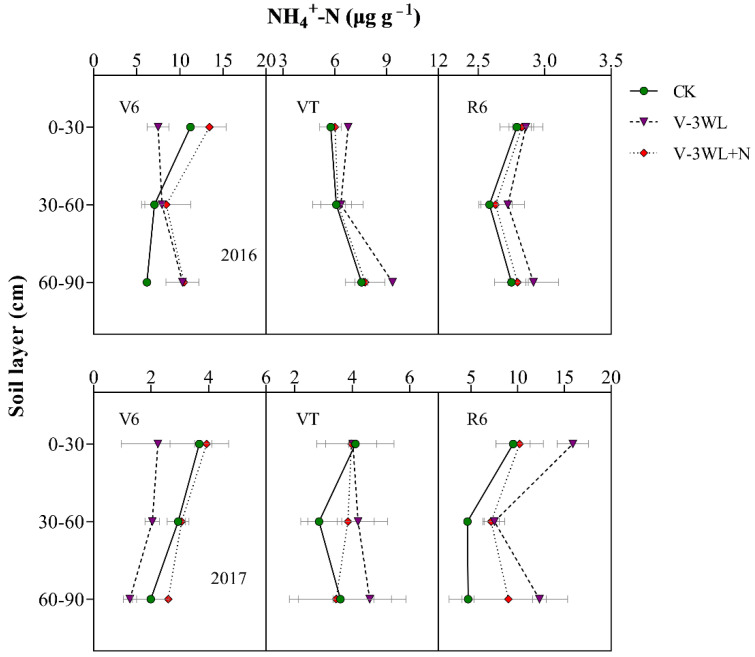
Effects of applying nitrapyrin on the soil NH_4_^+^-N content under a waterlogged field.

**Figure 5 plants-11-01983-f005:**
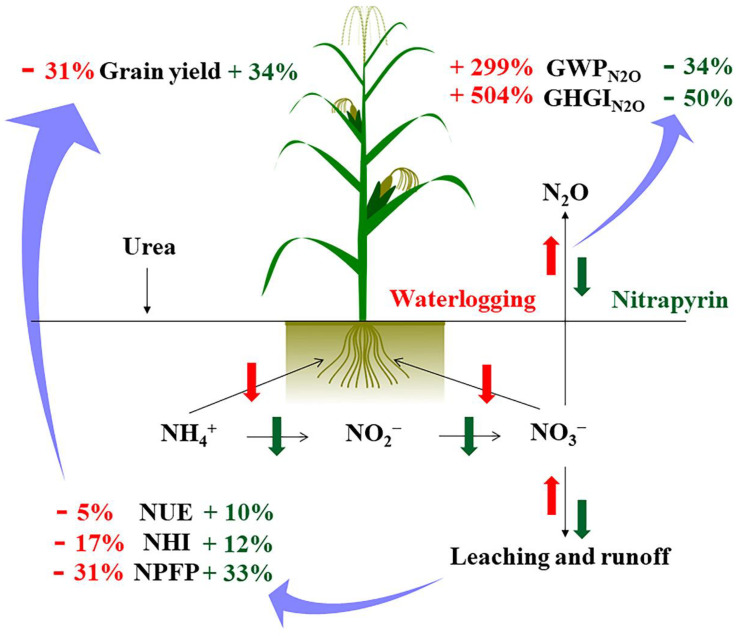
The influence process of applying nitrapyrin on soil N translocation under a waterlogged field. NUE, nitrogen use efficiency; NHI, nitrogen harvest index; NPFP, nitrogen partial factor productivity; GWP_N_2_O_, N_2_O warming potential; GHGI_N_2_O_, N_2_O greenhouse gas intensity.

**Table 1 plants-11-01983-t001:** Effects of applying nitrapyrin on grain yield and its components under a waterlogged field.

Year	Treatments	Harvest Ear Number (Ears ha^−^^1^)	Grains Number (Per Ear)	1000-Grains Weight (g)	Grain Yield (kg ha^−^^1^)
2016	V-3WL	61906c	447b	318c	8795c
	V-3WL+N	64407b	582a	341b	12803b
	CK	66908a	588a	351a	13821a
2017	V-3WL	62822b	490c	277b	9915c
	V-3WL+N	64038a	527b	309a	12141b
	CK	65724a	553a	311a	13159a

V-3WL, waterlogging treatment with urea application only; V-3WL+N, waterlogging treatment with urea and nitrapyrin application; CK, not waterlogged, but applied only urea. Means followed by a different letter (a,b,c) within a column differ at *p* < 0.05 by LSD test. The differences among treatments were calculated for each particular year.

**Table 2 plants-11-01983-t002:** Effects of applying nitrapyrin on the nitrogen accumulation and nitrogen efficiency under a waterlogged field.

Year	Treatment	Nitrogen Accumulation	Nitrogen Partial Factor Productivity	Nitrogen Use Efficiency	Nitrogen Harvest Index	Apparent Budget of Soil N
		(g p^−^^1^)	(NPFP, kg kg^−^^1^)	(NUE, kg kg^−^^1^)	(NHI)	(kg ha^−^^1^)
2016	V-3WL	2.75c	41.88c	51.66c	0.57c	33.56a
	V-3WL+N	3.05b	60.97b	59.34a	0.63b	9.79b
	CK	3.70a	65.81a	55.83b	0.69a	2.28c
2017	V-3WL	3.09c	47.21c	51.03b	0.52c	9.92a
	V-3WL+N	3.63b	57.81b	53.73a	0.59b	10.78a
	CK	3.97a	62.66a	52.69 a	0.63a	8.15b

V-3WL, waterlogging treatment with urea application only; V-3WL+N, waterlogging treatment with urea and nitrapyrin application; CK, not waterlogged, but applied only urea. Means followed by a different letter (a,b,c) within a column differ at *p* < 0.05 by LSD test. The differences among treatments were calculated for each particular year.

## Data Availability

The original contributions presented in the study are included in the article; further inquiries can be directed to the corresponding author/s.

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
