# Peer review of "Nitrapyrin Mitigates Nitrous Oxide Emissions, and Improves Maize Yield and Nitrogen Efficiency under Waterlogged Field"

_plants, 2022, doi:10.3390/plants11151983_

Round 1

Reviewer 1 Report

For improvement of the manuscript, revise with the following aspects:

1) Show the N-budget or mass balance  in the aspects of N-accumulation and N-use. (e.g., to take into account ammonium-N, nitrate-N, and N2O)

2) Discuss primary and secondary reasons for the differences in N2O emission and N-balance between Years 2016 and 2017.

3) Data presentation is somewhat unclear. Clarify.

Table 2 : What do the symbol a, b, c represent? 

All Equations : Do not use / for division or x for multiplication. Instead use the denominators and numerators.

All Figures : Do you mean 1 times standard error for representing the one side of error bars? To evade this confusion, use the 95% confidence interval, not standard error for representing error bars.

After the proper revision, the revised manuscript must be re-reviewed for potential publication.

Author Response

July 24 2022

Dear Reviewers

We would like to express our sincere thanks to you for your constructive and valuable comments, which are helpful for improving the quality of our manuscript. We have revised the text point by point in response to each comment including language, unit, formula, methodology, statistical analysis, and conclusion to improve the quality of the manuscript. The following shows that how and where the manuscript has been revised according to the comments.

Point 1: Show the N-budget or mass balance in the aspects of N-accumulation and N-use. (e.g., to take into account ammonium-N, nitrate-N, and N2O)

Response 1: Thanks for your valuable comments. We have made additions to the N-budget in the content of Table 2 and analyzed in the Methods, Results and Discussion section of the manuscript.

Line 154-156:Apparent N budget=∆N+NA-TN                                        (7)

where NA (kg N ha-1) is N applied, TN (kg ha-1) is total N uptake by plant, GN (kg ha-1) is grain N amount, ∆N (kg ha-1) is the change in soil inorganic N (including NH4+-N and NO3--N) before and after maize planting.

Table 2. Effects of applying nitrapyrin on nitrogen accumulation and nitrogen efficiency under waterlogged field.

Year

Treatment

Nitrogen accumulation

Nitrogen partial factor productivity

Nitrogen use efficiency

Nitrogen harvest index

Apparent budget of soil N

 (g p-1)

 (NPFP, kg kg–1)

 (NUE, kg kg–1)

(NHI)

(kg ha-1)

2016

V-3WL

2.75c

41.88c

51.66c

0.57c

33.56a

V-3WL+N

3.05b

60.97b

59.34a

0.63b

9.79b

CK

3.70a

65.81a

55.83b

0.69a

2.28c

2017

V-3WL

3.09c

47.21c

51.03b

0.52c

9.92a

V-3WL+N

3.63b

57.81b

53.73a

0.59b

10.78a

CK

3.97a

62.66a

52.69 a

0.63a

8.15b

Line 249-252: All treatments had a surplus of soil N, and the apparent surplus of soil N increased significantly after waterlogging. However, nitrapyrin application reduced the apparent soil N surplus in 2016 (no significant difference in 2017) (Table 2).

Line 324-332: The interannual N2O emission fluxes differences did not have a significant impact on the apparent budget of soil N (Table 2). The interannual differences appeared to be offset by soil inorganic N levels before N application in summer maize season. In addition, when calculating the apparent budget of soil N in this study, the mineralized and fixed amount of soil N were not included. The results showed that the N fertilization treatments had a surplus of N in summer maize season, and the waterlogging treatments had a greater risk of N loss. Nitrification inhibitors could reduce N leaching rates and improve N absorption and use efficiency in plants, thus mitigating waterlogging damages on N leaching and N use efficiency.

Point 2: Discuss primary and secondary reasons for the differences in N2O emission and N-balance between Years 2016 and 2017

Response 2: Thanks for your valuable comments. When we checked two years of N2O emission flux data, we found that the N2O emission flux data in 2016 was erroneously amplified due to a miscalculation. In view of this, we have recalculated and recreated the figures to correct the errors in the manuscript. This modification had no effect on the conclusions of this study. Furthermore, we have added a discussion of the reasons for the two-year difference in N2O emission fluxes to the Discussion section of the MS.

Line 272-274: In addition, sufficient mineral N content would promote the release of N2O emissions by changing the nitrification and denitrification rate of microorganisms, and even form “hot spots” of N2O emissions [4].

Line 285-285: Although the interaction of local temperature, soil and other environmental factors affect N2O emissions fluxes [46]. Compared with 2016, multi-day rainfall from June to July in 2017 may be the main reason for the difference in N2O emission fluxes between the two maize seasons.

Line 322-330: The interannual N2O emission fluxes differences did not have a significant impact on the apparent budget of soil N (Table 2). The interannual differences appeared to be offset by soil inorganic N levels before N application in summer maize season. In addition, when calculating the apparent budget of soil N in this study, the mineralized and fixed amount of soil N were not included. The results showed that the N fertilization treatments had a surplus of N in summer maize season, and the waterlogging treatments had a greater risk of N loss. Nitrification inhibitors could reduce N leaching rates and improve N absorption and use efficiency in plants, thus mitigating waterlogging damages on N leaching and N use efficiency.

Point 3: Data presentation is somewhat unclear. Clarify. Table 2 : What do the symbol a, b, c represent?

Response 3: Thanks for your valuable comments. We have made additions in the MS.

Line 258: Means followed by different letter (a,b,c) within a column differ at P < 0.05 by LSD test.

Point 4: All Equations : Do not use / for division or x for multiplication. Instead use the denominators and numerators

Response 4: Thanks for your valuable comments. We have corrected the formatting of these calculation formulas in the MS.

Point 5: All Figures : Do you mean 1 times standard error for representing the one side of error bars? To evade this confusion, use the 95% confidence interval, not standard error for representing error bars

Response 5: Thanks for your valuable comments. We have reworked all Figures in the MS. Error bars represent 95% confidence intervals.

Again, special thanks to you for your good comments. We tried our best to improve the manuscript and have made many corrections. These changes do not influence the content or framework of the paper. We sincerely appreciate the Editors/Reviewers’ work on this manuscript, and we hope that the revised version of the manuscript is now acceptable for publication in your journal. We will try our best to revise the manuscript till acceptable for publication in plants.

Sincerely,

Jiwang Zhang, Baizhao Ren

State Key Laboratory of Crop Biology and College of Agronomy, Shandong Agricultural University, Taian, Shandong 271018, China

E–mail: jwzhang@sdau.edu.cnï¼›renbaizhao@sina.com

Reviewer 2 Report

The article is correct to be published in the present form

Author Response

Thank you for your recognition of this article.